# Specificity Enhancement of Glutenase Bga1903 toward Celiac Disease-Eliciting Pro-Immunogenic Peptides via Active-Site Modification

**DOI:** 10.3390/ijms25010505

**Published:** 2023-12-29

**Authors:** Yu-You Liu, Rui-Ling Ye, Menghsiao Meng

**Affiliations:** Graduate Institute of Biotechnology, National Chung Hsing University, 250 Kuo-Kuang Rd., Taichung 40227, Taiwan; jojo89115@gmail.com (Y.-Y.L.); dti22830977p@gmail.com (R.-L.Y.)

**Keywords:** celiac disease, glutenase, gluten-free diet, gluten-derived pro-immunogenic peptides, oral enzyme therapy, enzyme active-site modification

## Abstract

Celiac disease is an autoimmune disease triggered by oral ingestion of gluten, with certain gluten residues resistant to digestive tract enzymes. Within the duodenum, the remaining peptides incite immunogenic responses, including the generation of autoantibodies and inflammation, leading to irreversible damage. Our previous exploration unveiled a glutenase called Bga1903 derived from the Gram-negative bacterium *Burkholderia gladioli*. The cleavage pattern of Bga1903 indicates its moderate ability to mitigate the toxicity of pro-immunogenic peptides. The crystal structure of Bga1903, along with the identification of subsites within its active site, was determined. To improve its substrate specificity toward prevalent motifs like QPQ within gluten peptides, the active site of Bga1903 underwent site-directed mutagenesis according to structural insights and enzymatic kinetics. Among the double-site mutants, E380Q/S387L exhibits an approximately 34-fold increase in its specificity constant toward the QPQ sequence, favoring glutamines at the P1 and P3 positions compared to the wild type. The increased specificity of E380Q/S387L not only enhances its ability to break down pro-immunogenic peptides but also positions this enzyme variant as a promising candidate for oral therapy for celiac disease.

## 1. Introduction

Celiac disease is an autoimmune disorder triggered by the consumption of gluten and is characterized by various gastrointestinal and extraintestinal symptoms [1]. Human leukocyte antigen (HLA)-DQ2 is associated with >90% of patients, while the remaining patients possess HLA-DQ8 [2]. It is noteworthy that additional unknown factors influence the development of the disease [3]. Celiac disease can be diagnosed initially by serological testing and confirmed via endoscopic biopsy that reveals villous atrophy and crypt hyperplasia in the small intestine [4,5]. Owing to the invasive nature of biopsy procedures, serologic tests, such as anti-transglutaminase 2 (TG2) antibodies and anti-endomysial antibody assays, are typically conducted in advance when there is suspicion of the disease [5].

The manifestation of the disease is primarily driven by gliadin, the ethanol-soluble component of wheat gluten [6]. The proline-rich composition renders gliadin partially resistant to proteolysis by human digestive enzymes. Some of the recalcitrant peptides, such as the 33-mer peptide from α2-gliadin and 26-mer peptide from γ5-gliadin, might enter the lamina propria of the duodenum via the transepithelial pathway [7,8], where these pro-immunogenic peptides are subsequently subjected to deamination by TG2, leading to the conversion of specific glutamine into glutamate residues [9]. Epitopes on the deaminated peptides are presented to gluten-specific CD4^+^ (cluster of differentiation 4) T cells by antigen-presenting cells possessing HLA-DQ2 and/or HLA-DQ8 [2]. The activated gluten-specific CD4^+^ T cells secrete inflammatory cytokines and stimulate specific B cells, leading to the production of anti-gliadin antibodies and anti-TG2 autoantibodies [10,11]. Since TG2 is implicated in cell-adhesive cytoskeletal cross-linking throughout the body [12], the assault on tissues by anti-TG2 autoantibodies would compromise intestinal barrier integrity, further increasing the influx of gluten peptides and exacerbating the systemic autoimmune response and malabsorption of nutrients [13].

At present, there is no cure for patients; therefore, a strict gluten-free diet is the only prescription to mitigate the onset of celiac disease. However, maintaining this diet for life is difficult because it is costly, inconvenient, and sometimes impossible due to inadvertent contamination during food processing [14,15]. Given the substantial burden that celiac disease imposes on public health and individual well-being, drugs with diverse mechanisms have been under development by pharmaceutical companies and academia [16,17]. Oral enzyme therapy has been suggested as one of the adjunctive treatments for celiac disease, involving the enzymatic hydrolysis of pro-immunogenic peptides derived from gluten within the stomach and duodenum [18,19,20].

In previous studies, we identified a serine peptidase, named Bga1903, with gliadin hydrolytic activity from *Burkholderia gladioli* [21]. Bga1903 has the capability of being secreted into the culture medium by *Escherichia coli* BL21(DE3). The structure of Bga1903 was determined through X-ray crystallography, and the architecture of the substrate-binding subsites was defined [22]. In the present study, an endeavor was made to further enhance its hydrolytic activity toward the 33- and 26-mer peptides by rationally modifying the S1 subsite of Bga1903 through mutagenesis. The alterations in catalytic constants were investigated to verify their respective effects.

## 2. Results

### 2.1. The Substrate-Binding Subsites of Bga1903

Bga1903 preferentially cut the bond at the carbonyl side of glutamine when it acted on the 26- and 33-mer peptides; nonetheless, it showed rather relaxed selectivity at the P1 residue, with a preference for lysine, followed by phenylalanine, leucine, and glutamine when bovine serum albumin served as the substrate [21]. These observations imply that elevating the catalytic specificity of Bga1903 toward pro-immunogenic peptides could be potentially achieved through strategic alterations of subsite S1 within the peptidyl-binding pocket. The crystal structure of Bga1903, highlighting residues constituting the S1–S3 subsites (Figure 1), indicates that the broader specificity for P1 residues may stem from the larger cavity of S1 compared to S2 and S3. An earlier study investigated the importance of residues E380 and S387, which line the wall of S1, in influencing the peptidase’s catalytic efficiency toward specific chromogenic peptidyl substrates [22]. The present study conducts a more detailed examination of these two residues, anticipating that tailoring S1 will result in changes in selectivity for acceptable P1 residues.

### 2.2. The Specific Activity of Bga1903 toward Various Chromogenic Tripeptidyl Substrates

To assess the substrate specificity of Bga1903, we employed several tripeptidyl chromogenic substrates, namely, Z-HPX-*p*NA and Z-QPQ-*p*NA (where Z stands for benzyloxycarbonyl, *p*NA for *p*-nitroanilide, and X for an amino acid). The design of Z-HPX-*p*NA was informed by Bga1903’s cleavage patterns on bovine serum albumins, revealing a preference for proline and histidine at the P2 and P3 positions, respectively [21]. Meanwhile, Z-QPQ-*p*NA represents a common motif present in the 26- and 33-mer peptides. The specific activity toward various tripeptidyl substrates is shown in Figure 2.

Within the Z-HPX-*p*NA group, the reactivity order is HPK >> HPQ > HPL > HPY ≈ HPF > HPE, affirming that the preferred order at the P1 position is K >> Q > L > Y ≈ F > E. Notably, Bga1903 exhibits reduced activity toward Z-QPQ-*p*NA compared to Z-HPQ-*p*NA, indicating a preference for histidine over glutamine at the P3 position. The high preference for Z-HPK-*p*NA and low preference for Z-HPE-*p*NA may be directly linked to the negatively charged potential in S1, particularly contributed by E380. Moreover, the S1 cavity appears to inadequately accommodate bulky hydrophobic groups, as evidenced by the disfavoring of substrates Z-HPY-*p*NA and Z-HPF-*p*NA.

### 2.3. Effect of E380 Substitutions on the Enzymatic Activity of Bga1903

To probe the function of E380 and explore the potential for altering the substrate specificity of Bga1903, E380 was substituted with eight alternatives, including four nonpolar amino acid residues (F, I, M, and W) and four polar ones (K, N, Q, and T). The catalytic constants of the mutants toward Z-QPQ-*p*NA, Z-HPQ-*p*NA, Z-HPL-*p*NA, and Z-HPK-*p*NA were then determined. The Michaelis constant (*K*_M_), which reflects the binding affinity to the ground-state substrate, and turnover number (*k*_cat_) are detailed in Table 1. The specificity constants (*k*_cat_/*K*_M_), providing insight into the binding affinity to the transition-state substrate, are illustrated in Figure 3. The enzymatic reactions utilizing the indicated substrates at varying concentrations were performed in triplicate. The representative plots of reaction rate versus substrate concentration of Bga1903 and E380-substituted mutants are displayed in Figure A1, Figure A2, Figure A3, Figure A4, Figure A5, Figure A6, Figure A7, Figure A8 and Figure A9. Insets show the corresponding Eadie–Hofstee plots.

In the context of wild-type Bga1903, *K*_M_ values range from 0.57 ± 0.07 to 3.49 ± 0.09 mM, while *k*_cat_/*K*_M_ values range from 30.95 ± 9.00 to 1.32 ± 0.04 mM^−1^·s^−1^. Notably, the peptidase exhibits a stronger binding affinity for Z-HPK-*p*NA, surpassing other tested substrates, in both ground-state and transition-state interactions. The substantial difference in *k*_cat_/*K*_M_ values indicates that the enhanced catalysis toward Z-HPK-*p*NA stems from an overwhelming affinity to the transition state of the substrate. Conversely, Z-QPQ-*p*NA emerges as the least preferable substrate, suggesting room for improvement if Bga1903 is to be considered for usage in celiac disease treatment. The significant difference in *k*_cat_/*K*_M_ values between Z-QPQ-*p*NA (1.32 ± 0.04 mM^−1^·s^−1^) and Z-HPQ-*p*NA (5.86 ± 0.18 mM^−1^·s^−1^), aligning with specific activity data, primarily arises from their distinct *k*_cat_ values. Accordingly, S3 also plays a pivotal role in the catalytic hydrolysis of the 26- and 33-mer peptides.

No detectable proteolytic activity toward Z-HPK-*p*NA was observed in any of the E380-substituted variants, underscoring the vital role of the negative charge associated with E380 in favoring lysine at the P1 position. Intriguingly, except for E380K, none of the mutants displays detectable proteolytic activity toward Z-HPL-*p*NA. After the introduction of E380K to Bga1903, the *k*_cat_/*K*_M_ value of E380K dropped from 3.10 ± 0.42 to 1.10 ± 0.08 mM^−1^·s^−1^ toward Z-HPL-*p*NA.

When Z-QPQ-*p*NA serves as the substrate, three mutations—E380W, E380K, and E380Q—significantly heighten the proteolytic activity of Bga1903. This enhancement is evident in the reduction in the *K*_M_ value from 3.16 ± 0.08 to 2.51 ± 0.01, 2.07 ± 0.07, and 0.88 ± 0.02 mM, accompanied by respective increases in the *k*_cat_/*K*_M_ value from 1.32 ± 0.04 to 4.17 ± 0.04, 7.20 ± 0.28, and 13.30 ± 0.58 mM^−1^·s^−1^.

Conversely, with Z-HPQ-*p*NA as the substrate, most mutants display either reduced or similar *k*_cat_/*K*_M_ values compared to the wild type. Notably, E380Q stands out by exhibiting the lowest *K*_M_ value, i.e., 0.69 ± 0.03 mM, among the variants. This change confers an approximately twofold increase in *k*_cat_/*K*_M_ on E380Q, compared to the wild type.

### 2.4. Cumulative Effect of S387L with Different E380 Mutations on the Enzymatic Activity

A prior investigation uncovered that the S387L mutation augmented the specific constant, *k*_cat_/*K*_M_, toward Z-HPQ-*p*NA and enhanced the breakdown of the 26-mer peptide. To assess whether the beneficial effects are accumulative, the impact of S387L mutation was re-evaluated in the current study on the background of E380K, E380W, and E380Q mutants as well as the wild-type enzyme. The resulting *k*_cat_ and *K*_M_ values are listed in Table 2, and the corresponding *k*_cat_/*K*_M_ values for each variant are illustrated in Figure 4. The representative plots of reaction rate versus substrate concentration of mutant S387L and various double-site mutants are displayed in Figure A10, Figure A11, Figure A12 and Figure A13. Insets show the corresponding Eadie–Hofstee plots.

Without interfering with the specificity constant toward Z-HPK-*p*NA, S387L mutation increases the *k*_cat_/*K*_M_ value by approximately 8-, 2-, and 3-fold toward Z-QPQ-*p*NA, Z-HPQ-*p*NA, and Z-HPL-*p*NA, respectively. It is worth noting that the increase in *k*_cat_/*K*_M_ is primarily driven by reductions in the *K*_M_ value rather than increases in the *k*_cat_ value in the case of Z-QPQ-*p*NA and Z-HPQ-*p*NA.

The mutant E380K/S387L has a *k*_cat_/*K*_M_ value of 12.04 ± 0.56 mM^−1^·s^−1^ toward Z-QPQ-*p*NA, which only increases slightly (~20%) compared to the mutant S387L. The *k*_cat_/*K*_M_ value reduced from 10.57 ± 0.96 to 5.03 ± 0.56 mM^−1^·s^−1^ if the substrate was replaced with Z-HPQ-*p*NA. From the perspective of eliminating QPQ motifs, the introduction of E380K substitution to the mutant S387L is not beneficial, since there is no obvious cumulative effect. However, it retains one of the properties from mutant E380K, in which no activity was detected utilizing Z-HPK-*p*NA and Z-HPL-*p*NA as substrates.

The mutant E380W/S387L exhibits a *k*_cat_/*K*_M_ value of 23.63 ± 0.80 mM^−1^·s^−1^ toward Z-QPQ-*p*NA, which is an approximately twofold and sixfold increase compared to mutants S387L and E380W, respectively. As for Z-HPQ-*p*NA, the *k*_cat_/*K*_M_ value increases from 10.57 ± 0.96 to 19.17 ± 1.25 when the E380W mutation is introduced to mutant S387L. The increase in *k*_cat_/*K*_M_ values for Z-QPQ-*p*NA and Z-HPQ-*p*NA is attributed to the respective *K*_M_ values, which decrease by nearly half when compared to their S387L counterparts. In contrast, no hydrolytic activity toward HPK-*p*NA and Z-HPL-*p*NA is detected, similar to the mutant E380K/S387L.

The mutant E380Q/S387L displays the highest *k*_cat_/*K*_M_ value toward Z-QPQ-*p*NA, i.e., 44.54 ± 2.53 mM^−1^·s^−1^, an approximately 3~4-fold increase when compared to either mutants E380Q or S387L. When the substrate is Z-HPQ-*p*NA, the *k*_cat_/*K*_M_ value changes to 46.22 ± 4.12, which is three to four times higher than those of E380Q and S387L, respectively. The exceptionally high *k*_cat_/*K*_M_ values for both Z-QPQ-*p*NA and Z-HPQ-*p*NA are attributed to the small *K*_M_ values among all variants. Although the mutant can hydrolyze Z-HPL-*p*NA, the *k*_cat_/*K*_M_ value is low compared to that of using Z-HPQ-*p*NA and Z-QPQ-*p*NA.

In summary, mutants E380W/S387L and E380Q/S387L have much higher specificity constant against Z-QPQ-*p*NA and Z-HPQ-*p*NA, demonstrating the combined impact of double-site mutations. Moreover, the pronounced preference for Z-HPK-*p*NA over Z-QPQ-*p*NA, originally seen in the wild type, has shifted in the opposite direction due to the double mutation E380Q/S387L.

### 2.5. Molecular Docking Models of Bga1903s and QPQ Peptides

To elucidate the underlying mechanism responsible for the increased specificity constant resulting from double mutations (E380W/S387L or E380Q/S387L), computational docking models of peptidase Bga1903 with a QPQ-peptide substrate were constructed using HADDOCK v.2.4 software. Interactions between the QPQ peptide and different versions of Bga1903 were plotted according to the simulation provided by LigPlot+ software v.2.2.8, as depicted in Figure 5.

In the wild-type Bga1903, as depicted in Figure 5A, a charged hydrogen bond is formed between the OE2 atom of E380 and the NE2 atom of glutamine at the P1 position. Additional hydrogen bonds are also established between the backbones of the QPQ peptide and the P3 subsites (S348 and G350), acting as anchor points to further stabilize the QPQ substrate.

In the case of mutant S387L, as illustrated in Figure 5B, the presence of L387 pushes the side chain of glutamine (P1) closer to the backbone of G378-N379-E380; consequently, two more hydrogen bonds are formed between the P1 residue and the subsites when compared to the wild-type Bga1903. This may account for the improved binding affinity for the QPQ substrate in both the ground state and transition state.

For mutant E380W/S387L, as shown in Figure 5C, there is a noteworthy rotation in the side chain of glutamine (P1) when compared to the wild-type scenario. This rotation results in the formation of two hydrogen bonds. One of these bonds is directed toward the NE1 atom on the side chain of W380, while the other connects with the N atom on the backbone of N379. Additionally, the carboxyl end of glutamine (P1) establishes two more hydrogen bonds, one with the ND2 atom of N379 and the other with the N atom on the backbone of S449. These interactions further enhance the substrate’s affinity.

As for mutant E380Q/S387L, as presented in Figure 5D, the rotation of the glutamine side chain (P1) is similar to that of the E380W/S387L mutant, and a hydrogen bond forms between Q380 and the glutamine at the P1 position of the QPQ substrate. An additional hydrogen bond is formed between Q262 and the P3 residue of the QPQ substrate, compared to the E380W/S387L scenario. This could explain why mutant E380Q/S387L exhibits the highest specificity constant toward the QPQ substrate among various mutated versions of Bga1903.

### 2.6. Hydrolysis of Pro-Immunogenic Peptides by Mutant E380W/S387L and E380Q/S387L

To verify their hydrolytic activity toward the 26-mer or 33-mer peptide, wild-type Bga1903, mutant E380W/S387L, and mutant E380Q/S387L were incubated with the peptide at pH 6.0. After incubation at 37 °C for 2 h, the hydrolysate was subjected to RP-HPLC for analysis, as depicted in Figure 6. Overall, the 26-mer peptide is more susceptible to hydrolysis than the 33-mer peptide. Both the double-mutant variants exhibit stronger proteolytic activity than the wild type toward both peptides. Particularly noteworthy is the superior efficiency of mutant E380Q/S387L in hydrolyzing the 33-mer peptide as compared to the wild type and mutant E380W/S387L.

## 3. Discussion

Given their potential as supporting therapeutic agents for celiac disease, glutenases that can effectively degrade toxic peptides derived from gluten have been extensively investigated [18,23]. A couple of them have even undergone clinical trials to assess their safety and efficacy. It is worth mentioning that the glutenases under development do not intend to serve as a full substitute for the gluten-free diet. First, the substantial presence of gluten in common diets poses a challenge for relying solely on glutenase treatment, given the considerable dosage needed to achieve comprehensive hydrolysis. Second, many other gluten-derived celiac-disease-inducing epitopes, rather than those found in 26- and 33-mer peptides, may persist after the administration of the glutenases under development. Nevertheless, the oral administration of glutenases is still expected to support individuals adhering to a gluten-free diet by easing symptoms or providing a safeguard against unintentional gluten consumption.

Glutenases, which belong to the aspartate, cysteine, glutamate, or serine peptidase family, have been discovered in various sources, including plants, fungi, and bacteria. [21]. Nepenthesin, an aspartate protease, and neprosin, a glutamate protease, were initially isolated from the fluid within the pitcher leaf of tropical carnivorous plant *Nepenthes × ventrata* [24,25,26]. These enzymes can break down the 33-mer peptide at pH 2.5, showing a preference for proline at the P1 position. Moreover, they have demonstrated the ability to prevent gliadin-induced inflammation in gluten-sensitive NOD/DQ8 transgenic mice.

RmuAP1, identified as an aspartate protease from the red yeast *Rhodotorula mucilaginosa*, exhibits hydrolysis activity toward 26- and 33-mer peptides by cleaving the peptide bond at the carbonyl side of glutamine over a pH range of 3.0–6.0 [27]. AN-PEP, a serine endopeptidase of the S28 family isolated from *Aspergillus niger*, shows a preference for cleaving the peptide bond after proline [28]. Notably, AN-PEP has been marketed both as a dietary supplement (Tolerase G) claimed to ease the symptoms of gluten-related diseases and as a food additive (Brewers Clarex) designed to remove chill haze during beer brewing [29,30].

Rmep is a serine peptidase of the S8 family, produced by *Rothia mucilaginosa*, a commensal bacterium naturally occurring in the human oral cavity [31]. Rmep exhibits the capability to preferentially degrade the 33-mer peptide at the bonds after glutamine and tyrosine, with optimal activity observed at pH 9.0. Another S8 protease with gluten-degrading activity is Subtilisin Carlsberg, produced by *Bacillus licheniformis* [32]. However, the performance of Subtilisin Carlsberg is compromised at acidic conditions, and it is prone to autolysis easily. To mitigate these issues, modifications such as PEGylation and polylactic glycolic acid (PLGA) microencapsulation have been applied to Subtilisin Carlsberg. These adjustments render the enzyme less susceptible to acidic exposure or autolysis, thereby enhancing its efficiency in detoxifying peptides derived from gluten [33].

EP-B2 is a cysteine endopeptidase discovered in germinating barley seeds. It can hydrolyze the 33-mer peptide with a preference for peptide bonds after glutamine [34,35]. EP-B2, when presented as the zymogen form, exhibits optimal activity at pH 4.5. On the other hand, SC-PEP, a serine prolyl endopeptidase from *Sphigomonas capsulate*, belonging to the S9 family, favors proline over glutamine in the P1 position, with optimal pH values ranging from 6 to 7 [36]. A combined approach involving recombinant EP-B2 and SC-PEP, known as Latiglutenase (formerly ALV003 or IMGX003), has undergone extensive investigation in human clinical trials, marking it as the most thoroughly studied intervention to date [37,38,39,40,41]. The phase II clinical trials for this drug yielded cautiously optimistic results, suggesting promising therapeutic potential [40,41]. Theoretically, EP-B2 and SC-PEP complement each other not only in their substrate preferences but also in their action sites, spanning both the stomach and duodenum where the proteolytic reactions take place.

Kuma-062 (also called TAK-062), an acid-tolerant S53 peptidase derived from kumamolisin of *Alicyclobacillus sendaiensis* through an iterative engineering process, efficiently eliminates the gluten-derived peptides at pH 4.0, with a preference for glutamine at the P1 position [42,43,44]. In the phase I trial for the dose escalation study, Kuma-062 demonstrated high effectiveness in degrading gluten within standardized meals, achieving a rate of over 97% [44]. Currently, in the recruitment stage of phase II trials, the impact of Kuma-062 on patients of celiac disease will be revealed in the future.

In the present study, site-directed mutagenesis was employed to modify the S1 subsite of Bga1903. The importance of E380 in determining the peptidase’s preference for lysine at the P1 position was established. The peptidase’s activity toward Z-HPK-*p*NA is not supported by any substitution for E380. However, a notable improvement in the peptidase’s activity toward Z-QPQ-*p*NA is observed upon the replacement of E380 with glutamine. When compared to the wild type, the substrate preference of the E380Q mutant undergoes a reversal from the HPK peptide, which is the most favored substrate by the wild type, to the QPQ and HPQ peptides, which are disfavored by the wild type. The introduction of the S387L mutation further heightens the peptidase’s preference for QPQ (with the *k*_cat_/*K*_M_ value of 44.54 ± 2.53 mM^−1^·s^−1^) and HPQ (with the *k*_cat_/*K*_M_ value of 46.22 ± 4.12 mM^−1^·s^−1^). The approximately 34-fold increase in the *k*_cat_/*K*_M_ value toward Z-QPQ-*p*NA should account for the enhanced activity of E380Q/S387L in the hydrolysis of 26- and 33-mer peptides.

In summary, reshaping the catalytic subsites of Bga1903 has significantly enhanced its activity for the removal of gluten-derived toxic peptides. With its preference for glutamine and optimum activity at pH 6.0–7.0, the double-point mutant E380Q/S387L holds promise as a key component in a prospective glutenase formula for treating celiac disease, especially when combined with other peptidases possessing complementary properties, e.g., favoring proline at the P1 position and acting in more acidic conditions.

## 4. Materials and Methods

### 4.1. Chemicals and Reagents

Chromogenic peptidyl substrates, including Z-HPK-*p*NA, Z-HPQ-*p*NA, Z-HPL-*p*NA, Z-HPY-*p*NA, Z-HPF-*p*NA, Z-HPE-*p*NA, and Z-QPQ-*p*NA were synthesized by Kelowna International Scientific (Taipei, Taiwan). Q5 site-directed mutagenesis kit was purchased from New England Biolabs (Ipswich, MA, USA). Primers for PCR used in mutagenesis were synthesized by Genomics (New Taipei, Taiwan), and listed in Table 3. The 26- and 33-mer immunogenic peptides were chemically synthesized by Mission Biotech (Taipei, Taiwan).

### 4.2. E. coli Strains and Site-Directed Mutagenesis

*E. coli* BL21(DE3) pLysS was purchased from Merck (Rahway, NJ, USA), and utilized as the host for recombinant peptidase production. The process of obtaining plasmid pET-*Bga1903*^+^ was mentioned previously [21]. In brief, the *Bga1903* gene cloned from *B. gladioli* was inserted into plasmid pET-Duet1, and the resulting plasmid pET-*Bga1903* was chemically resynthesized to a codon-optimized version, pET-*Bga1903^+^*. The substitutions of amino acids were performed through PCR-based site-directed mutagenesis on plasmid pET-*Bga1903^+^* utilizing a Q5 site-directed mutagenesis kit.

### 4.3. Protein Expression and Purification

Recombinant Bga1903 was produced and purified as described previously [21]. In brief, *E. coli* BL21(DE3) pLysS was cultured overnight with Lysogeny broth (LB, containing 10 g/L tryptone, 5 g/L yeast extract, and 10 g/L NaCl). Fresh LB medium containing 100 μg/mL ampicillin inside a shake flask was inoculated with overnight bacterial culture, and placed inside a shaking incubator at 37 °C at 200 rpm until the OD_600_ reached approximately 1.0. Administrated with 1 mM final concentration of isopropyl β-D-1-thiogalactopyranoside, the cell culture was incubated for 18 h at 28 °C at 200 rpm. To harvest secreted Bga1903 within the medium, the bacterial broth was centrifuged at 10,000× *g* for 15 min at 4 °C, and the cell pellet was discarded afterward. Bga1903 inside the culture medium was harvested and subsequently purified by using a column packed with Qiagen Ni-NTA agarose resin (Venlo, The Netherlands). The purified protein was obtained by concentrating elution fractions using a 10 kDa Amicon ultra-4 centrifugal filter unit (Merck, Darmstadt, Germany), and preserved in the buffer containing 20 mM Tris [pH 8.0] and 500 mM NaCl. The concentration of the purified protein was measured using Bradford protein assay reagent (Thermo Fisher Scientific, Waltham, MA, USA) with bovine serum albumin as the standard.

### 4.4. Enzyme Activity Assay Utilizing Chromogenic Peptidyl Substrates

The enzymatic activity toward various chromogenic peptidyl substrates was measured according to the release of *p*-nitroaniline from the substrates. One activity unit (U) was defined as the release of 1 nmol *p*-nitroaniline per second at the indicated conditions. Unless specified, the reaction containing 0.5 mM peptidyl substrates, 12.5 μg/mL of the purified enzyme, 1.5% (*v*/*v*) Triton X-100, and 40 mM citrate-phosphate buffer (pH 6.0) was performed at 37 °C. The catalytic rate was calculated following the time-course increment of optical density at 405 nm and based on the molar extinction coefficient of *p*-nitroaniline, 9960 M^−1^·cm^−1^. For steady-state kinetics, the concentration of chromogenic peptidyl substrates was subject to change when the reaction was performed in triplicate. Eadie–Hofstee plots were employed for the determination of *K*_M_ (Michaelis constant) and *V*_max_ (turnover number) values concerning different substrates. Based on the Michaelis–Menten equation, the Eadie–Hofstee plot is illustrated according to the equation: *v* = *V*_max_ − (*K*_M_·*v*/[S]), where *v* and [S] represent the release rate of *p*-nitroaniline and the concentration of the substrate, respectively. The *k*_cat_ values are derived using the equation: *V*_max_ = *k*_cat_ × [E]_t_, where [E]_t_ represents the molar concentration of Bga1903. Due to the limited solubility of the chromogenic peptidyl substrates, the value of *k*_cat_/*K*_M_ of E380T was obtained from the plot of initial velocity, which is in the first-order reaction range, versus substrate concentration based on the equation: *v* = *k*_cat_/*K*_M_·[E][S].

### 4.5. Computational Evaluation of the Interaction between Bga1903s and QPQ Peptide

The tertiary structure determination of Bga1903 by X-ray crystallography was described previously [22]. Bga1903′s atomic coordinates and structure factors were deposited in the Protein Data Bank with PDB ID code 7W2A. Interactions of the representative motif of 26-mer peptide, QPQ, and mutated variants of Bga1903 were built according to the suggestion of FoldX software v.2.5.4 [45], and their computational dockings were performed with HADDOCK2.4 website [46,47]. The interactions between QPQ peptides and Bga1903 molecules were predicted by Ligplot+ v.2.2.8 [48].

### 4.6. Hydrolysis of the 26- and 33-mer Pro-Immunogenic Peptides

The hydrolysis of the 26- and 33-mer pro-immunogenic peptides was carried out by incubating the peptide (1 mg/mL) and the purified peptidase (50 μg/mL) in 50 mM citrate-phosphate buffer, pH 6.0 at 37 °C for 2 h, followed by incubation at 95 °C for 5 min to deactivate the peptidase. The degree of the peptide degradation was analyzed with a Gilson HPLC system (Middleton, WI, USA) using a C18 column (Ascentis Express 25 cm × 4.6 mm, 5 µm, Supelco, Bellefonte, PA, USA). After sample injection, the C18 column was subjected to H_2_O with 0.1% (*v*/*v*) trifluoroacetic acid for 5 min and then proceeded with linear gradient elution mode from 0 to 80% acetonitrile with 0.1% (*v*/*v*) trifluoroacetic acid for 20 min. The flow rate was set at a constant 1 mL/min, and the elution was monitored at 230 nm.

## Figures and Tables

**Figure 1 ijms-25-00505-f001:**
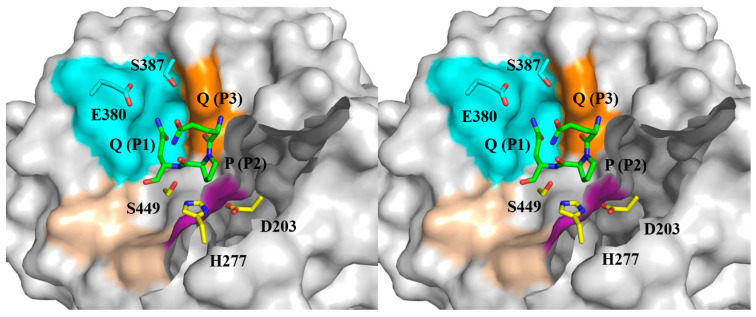
The docking model featuring the active site of Bga1903 complexed with QPQ tripeptide in stereo view. The coloring scheme designates the surface of the S1, S2, S3, and S1′ sites as cyan, magenta, orange, and wheat, respectively. Residues E380 and S387, targeted for mutagenesis in the present study, are colored elementally (C: cyan; N: blue; O: red). The QPQ tripeptide containing the corresponding P1–P3 residues to the subsites S1–S3 is also colored elementally (C: green; N: blue; O: red). The catalytic triad composed of D203, H277, and S449 is colored by elements (C: yellow; N: blue; O: red).

**Figure 2 ijms-25-00505-f002:**
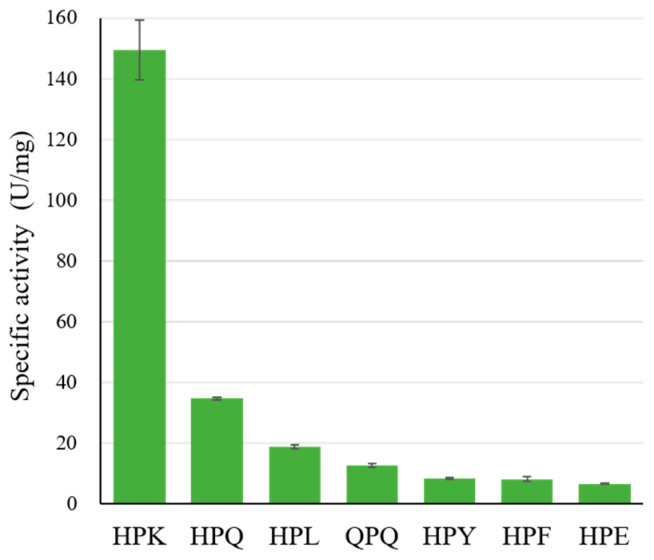
The specific activity of Bga1903 toward chromogenic tripeptidyl substrates. The reactions were performed in triplicate at 37 °C, pH 6.0. The activity rate was calculated based on the release of *p*-nitroaniline as described in Materials and Methods. Error bars denote the standard deviation of mean values from three replicates.

**Figure 3 ijms-25-00505-f003:**
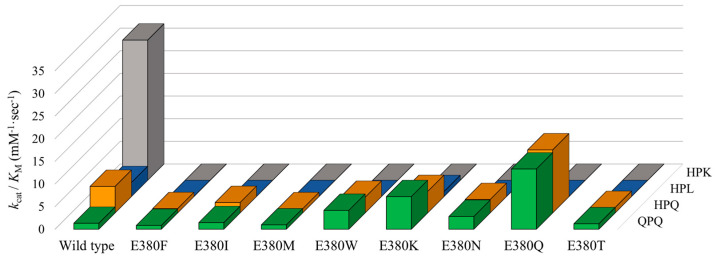
The specificity constants of Bga1903 and E380-substituted mutants toward selected substrates. The specificity constants (*k*_cat_/*K*_M_) toward Z-QPQ-*p*NA (green), Z-HPQ-*p*NA (orange), Z-HPL-*p*NA (blue), and Z-HPK-*p*NA (gray) are derived from the values of *k*_cat_ and *K*_M_ presented in Table 1.

**Figure 4 ijms-25-00505-f004:**
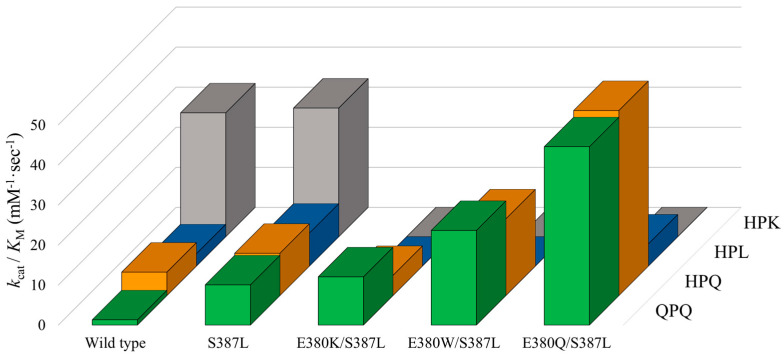
Cumulative effects of mutations at S387 and E380. The specificity constants, *k*_cat_/*K*_M_, of Bga1903 and the indicated variants toward selected peptidyl substrates, including Z-QPQ-*p*NA (green), Z-HPQ-*p*NA (orange), Z-HPL-*p*NA (blue), or Z-HPK-*p*NA (gray), are derived from the values of *k*_cat_ and *K*_M_ presented in Table 2.

**Figure 5 ijms-25-00505-f005:**
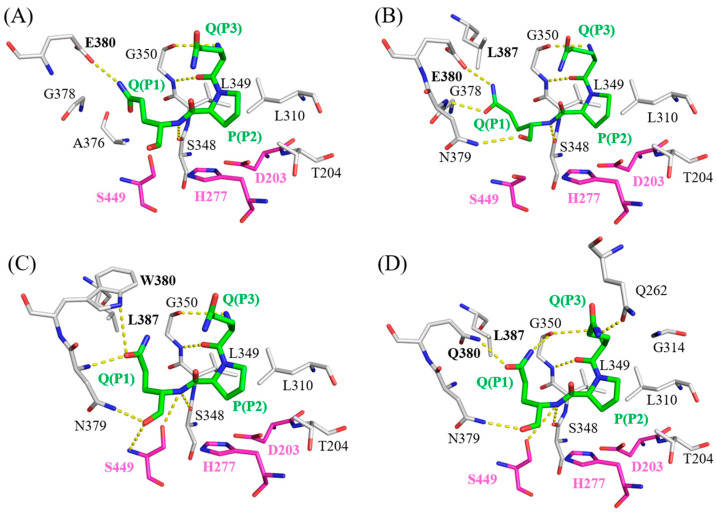
The molecular docking models of the QPQ peptides with different variants of Bga1903s, including (**A**) wild-type Bga1903, (**B**) mutant S387L, (**C**) mutant E380W/S387L, and (**D**) mutant E380Q/S387L. The QPQ peptides located at S1–S3 positions are colored elementally (C: green; N: blue; O: red), and spatially neighboring residues from the peptidase are colored by elements (C: white; N: blue; O: red). The catalytic triad (D203, H277, S449) is colored by elements (C: magenta; N: blue; O: red). H-bonds (hydrogen bonds) between N atoms and C atoms are illustrated as yellow-dotted lines.

**Figure 6 ijms-25-00505-f006:**
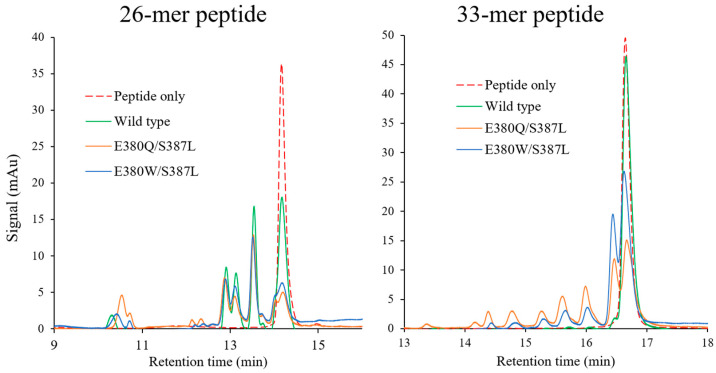
Hydrolysis of pro-immunogenic peptides by wild-type Bga1903, mutant E380Q/S387L, and mutant E380W/S387L. The 26-mer or 33-mer peptide (1 mg/mL) was incubated with or without the peptidase (50 μg/mL) at pH 6.0 for 2 h, and the digested fragments were analyzed by RP-HPLC.

**Table 1 ijms-25-00505-t001:** Mutational effects of E380 substitution on the kinetic constants of selected peptidyl substrates.

Variants	QPQ		HPQ		HPL		HPK	
	*k*_cat_ ^a^	*K* _M_	*k* _cat_	*K* _M_	*k* _cat_	*K* _M_	*k* _cat_	*K* _M_
	(s^−1^)	(mM)	(s^−1^)	(mM)	(s^−1^)	(mM)	(s^−1^)	(mM)
Wild type	4.17 ± 0.06	3.16 ± 0.08	20.43 ± 0.37	3.49 ± 0.09	1.77 ± 0.12	0.57 ± 0.07	49.90 ± 7.44	1.61 ± 0.40
E380F	3.69 ± 0.05	4.34 ± 0.06	2.99 ± 0.09	3.17 ± 0.21	BDL ^b^	BDL	BDL	BDL
E380I	7.54 ± 0.45	5.16 ± 0.30	5.09 ± 0.14	2.20 ± 0.10	BDL	BDL	BDL	BDL
E380M	4.97 ± 0.38	5.18 ± 0.30	6.19 ± 0.44	6.72 ± 0.58	BDL	BDL	BDL	BDL
E380W	10.49 ± 0.09	2.51 ± 0.01	9.13 ± 0.40	2.65 ± 0.20	BDL	BDL	BDL	BDL
E380K	14.86 ± 0.28	2.07 ± 0.07	10.42 ± 0.21	2.21 ± 0.02	1.00 ± 0.01	0.90 ± 0.07	BDL	BDL
E380N	9.93 ± 0.12	3.52 ± 0.01	10.50 ± 0.46	3.55 ± 0.11	BDL	BDL	BDL	BDL
E380Q	11.75 ± 0.41	0.88 ± 0.02	9.60 ± 0.30	0.69 ± 0.03	BDL	BDL	BDL	BDL
E380T	UD ^c^	UD	UD	UD	BDL	BDL	BDL	BDL

a. The kinetic constants *k*_cat_ and *K*_M_ are determined based on the Eadie–Hofstee plot. Each reaction condition was performed in triplicate, and the mean value ± SD (standard deviation) of the calculated *k*_cat_ and *K*_M_ are presented. b. BDL stands for below detection limit. c. UD stands for undetermined, indicating individual *k*_cat_ and *K*_M_ values cannot be accurately obtained because the used substrate concentrations are well below the expected *K*_M_.

**Table 2 ijms-25-00505-t002:** Mutational effects of double mutations at E380 and S387 on selected peptidyl substrates.

Variants	QPQ		HPQ		HPL		HPK	
	*k*_cat_ ^a^	*K* _M_	*k* _cat_	*K* _M_	*k* _cat_	*K* _M_	*k* _cat_	*K* _M_
	(s^−1^)	(mM)	(s^−1^)	(mM)	(s^−1^)	(mM)	(s^−1^)	(mM)
Wild type	4.17 ± 0.06	3.16 ± 0.08	20.43 ± 0.37	3.49 ± 0.09	1.77 ± 0.12	0.57 ± 0.07	49.90 ± 7.44	1.61 ± 0.40
S387L	10.17 ± 0.15	1.01 ± 0.05	9.68 ± 0.42	0.92 ± 0.07	7.39 ± 0.14	0.90 ± 0.05	22.50 ± 1.04	0.70 ± 0.02
E380K/S387L	7.20 ± 0.09	0.60 ± 0.03	5.83 ± 0.33	1.16 ± 0.11	BDL ^b^	BDL	BDL	BDL
E380W/S387L	8.92 ± 0.06	0.38 ± 0.01	7.42 ± 0.10	0.39 ± 0.03	BDL	BDL	BDL	BDL
E380Q/S387L	10.36 ± 0.16	0.23 ± 0.01	8.67 ± 0.11	0.19 ± 0.02	5.47 ± 0.79	0.99 ± 0.19	BDL	BDL

a. The kinetic constants *k*_cat_ and *K*_M_ are determined based on the Eadie–Hofstee plot. Each reaction condition was performed in triplicate, and the mean value ± SD (standard deviation) of the calculated *k*_cat_ and *K*_M_ are presented. b. BDL stands for below detection limit.

**Table 3 ijms-25-00505-t003:** Primers utilized in site-directed mutagenesis.

Primer Name	Sequence (5′→3′)
1903^+^ E380-R	GGCCACAACCACGGTGGCACCTTTGGC
1903^+^ E380I-F	GCCGGTAATATTGCGAGCCCGGTTAGC
1903^+^ E380K-F	GCCGGTAATAAAGCGAGCCCGGTTAGC
1903^+^ E380M-F	GCCGGTAATATGGCGAGCCCGGTTAGC
1903^+^ E380N-F	GCCGGTAATAACGCGAGCCCGGTTAGC
1903^+^ E380Q-F	GCCGGTAATCAGGCGAGCCCGGTTAGC
1903^+^ E380T-F	GCCGGTAATACCGCGAGCCCGGTTAGC
1903^+^ E380W-F	GCCGGTAATTGGGCGAGCCCGGTTAGC
1903^+^ E380Y-F	GCCGGTAATTATGCGAGCCCGGTTAGC

## Data Availability

Data available within the article.

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
