# Peer review of "Specificity Enhancement of Glutenase Bga1903 toward Celiac Disease-Eliciting Pro-Immunogenic Peptides via Active-Site Modification"

_ijms, 2023, doi:10.3390/ijms25010505_

Round 1
Reviewer 1 Report
Comments and Suggestions for Authors
Here are my comments on the glutenase manuscript:
- The discussion chapter is unusually short while the results are several pages. Lots of the results are now discussed in results and the discussion chapter loses its sginificance.
- authors should discuss the potential side-effects of this treatment in the discussion.
- Also, is the treatment focused adjunct to GFD or replace it. What is the target group of patients?
Author Response
- The discussion chapter is unusually short while the results are several pages. Lots of the results are now discussed in results and the discussion chapter loses its significance.
Respond: The discussion chapter has been revised to provide more information associated with other glutenases that are currently under development.
- Authors should discuss the potential side-effects of this treatment in the discussion. Also, is the treatment focused adjunct to GFD or replace it. What is the target group of patients?
Respond: The oral therapeutic glutenases are not meant to completely replace GFD. This information is added in lines 261-267. The glutenases are applied to patients who follow gluten-free regimes to alleviate the reoccurring symptoms or prevent them.
Reviewer 2 Report
Comments and Suggestions for Authors
This manuscript is a continuation of previous articles published by the same authors (refs 24 & 25) about glutenase Bga1903. This enzyme is able to degrade gluten immunogenic peptides. Based upon their previous data on wildtype enzyme, including its X-ray structure, in the current manuscript they have performed an extense study of active-site mutagenesis (residues E380 and S387) and enzyme characterization of the mutants (kcat, KM and kcat/KM) to search for more efficient Bga1903 variants. For the characterization, they have used several model tripeptide substrates. The best result was obtained with the double mutant E380Q/S387L, which displays an alteration of substrate preference with respect to the wildtype enzyme and a 38-fold increased catalytic efficiency on the QPQ tripeptide. This result is rationalized by means of molecular docking of the QPQ peptide to the wildtype structure of Bga1903 and to the predicted structure of enzyme mutants.
Since this tripeptide is a prevalent motif present in the 26- and 33-mer immunogenic peptides of gluten, the activity on these larger peptides was also tested with good results favoring E380Q/S387L mutant. The authors suggest that the E380Q/S387L mutant of Bga1903 is a promising candidate for the therapy of celiac disease.
I find the manuscript interesting, and I would like to make several suggestions.
1.- The saturation kinetics experiments are essential to the quality of the work. The manuscript contains lots of kcat and KM values in tables with error values, but the errors are not explained.
2.- In section 4.4, please explain how the kinetic parameters were calculated.
3.- The saturation curves from where kcat and KM values have been derived are not shown. I suggest to prepare a new figure with the most interesting ones (at least, wt and E380Q/S387L mutant). The manuscript will be also improved if all the saturation curves are shown in a supplemental file.
4.- Line 18 and line 294 give contradictory data: respectively, 38-fold and 44-fold increase for the same comparison.
5.- In the Discussion, the authors should be more explicit in comparing the characteristics of the E380Q/S387L mutant with those of other gluten-degrading proteases, particular with those in clinical/trial use.
6.- Lines 297-299, please state what complementary properties should be displayed by any other protease (real or desirable) to complement the action of the E380Q/S387L mutant.
Author Response
- The saturation kinetics experiments are essential to the quality of the work. The manuscript contains lots of kcat and KM values in tables with error values, but the errors are not explained.
Response: The explanation of error values is added in line 125 at the bottom of Table 1 and in lines 165-167 at the bottom of Table 2. The kcat values and KM values consist of individual mean values and standard deviations.
- In section 4.4, please explain how the kinetic parameters were calculated.
Response: The methods for the calculation of kinetic parameters according to Michaelis–Menten equation is shown in lines 376-380.
- The saturation curves from where kcat and KM values have been derived are not shown. I suggest to prepare a new figure with the most interesting ones (at least, wt and E380Q/S387L mutant). The manuscript will be also improved if all the saturation curves are shown in a supplemental file.
Response: The plots of initial rates of releasing p-nitroaniline under different concentrations of peptidyl substrates utilizing wild-type Bga1903 or its variations and their corresponding double reciprocal plots are shown in the newly added Appendix as Figures A1-A13 (pages 12-17).
- Line 18 and line 294 give contradictory data: respectively, 38-fold and 44-fold increase for the same comparison.
Response: Both of them are correctly revised as 39-fold increase, as depicted in line 18 and line 324 (formally termed line 292).
- In the Discussion, the authors should be more explicit in comparing the characteristics of the E380Q/S387L mutant with those of other gluten-degrading proteases, particular with those in clinical/trial use.
Response: The discussion chapter has been revised to provide more information associated with other glutenases that are currently under development. The comparison with Latiglutenase (a cocktail formula of EP-B2 and SC-PEP) and Kuma-062 is listed in lines 270-313.
- Lines 297-299, please state what complementary properties should be displayed by any other protease (real or desirable) to complement the action of the E380Q/S387L mutant.
Response: The complementary properties of other glutenases include with optimal pH at acidic conditions, or the P1 preference for proline, as described in lines 328-332.
Reviewer 3 Report
Comments and Suggestions for Authors
In the present basic science study Liu et al modified by mutagenesis a bacterial-derived glutenase (Bga1903) in order to increase the hydrolytic activity against gluten peptides.
I think that the study was well planned and the way Authors provided and described the results is effective and well readable. My only comment is that Authors should better describe possible limitations of their study in the Discussion paragraph. For example, even though 33-mer is recognized as one of the most important peptides eliciting autoimmune response in CD, it is not the only one. This may explain why peptidases have not shown so far very exciting results in human trials. Herein, indeed, Authors did not test proteolytic activity against other gluten sequences.
Author Response
In the present basic science study Liu et al modified by mutagenesis a bacterial-derived glutenase (Bga1903) in order to increase the hydrolytic activity against gluten peptides.
I think that the study was well planned and the way Authors provided and described the results is effective and well readable. My only comment is that Authors should better describe possible limitations of their study in the Discussion paragraph. For example, even though 33-mer is recognized as one of the most important peptides eliciting autoimmune response in CD, it is not the only one. This may explain why peptidases have not shown so far very exciting results in human trials. Herein, indeed, Authors did not test proteolytic activity against other gluten sequences.
Response: We appreciate the insight the reviewer provided us. Even though 33-mer and 26-mer peptides are the major factors, other peptide motifs may be overlooked. Another limitation is that glutenases will still require patients to follow the gluten-free diet since normal diets contain too much amount of gluten. These two limitations of glutenases are listed in lines 261-267.
Reviewer 4 Report
Comments and Suggestions for Authors
The manuscript title Specificity Enhancement of Glutenase Bga1903 toward Celiac 2 Disease-Eliciting Pro-immunogenic Peptides via 3 Active-Site Modification
The provided research article discusses celiac disease, an autoimmune condition triggered by the ingestion of gluten, which results in immune responses, autoantibody generation, and inflammation in the duodenum. The focus is on Bga1903, a glutenase derived from Burkholderia gladioli, previously identified for its ability to cleave gluten peptides. The crystal structure of Bga1903 is determined, revealing its active site and subsites. The text then describes site-directed mutagenesis to enhance Bga1903's substrate specificity, particularly targeting prevalent motifs like QPQ within gluten peptides. The double-site mutant E380Q/S387L emerges as a promising candidate, exhibiting a significant increase in specificity for the QPQ sequence compared to the wild type. This heightened specificity enhances the hydrolytic activity against pro-immunogenic peptides, positioning E380Q/S387L as a potential oral therapy enzyme for celiac disease
In the current study, the researchers aimed to build upon previous findings related to the serine peptidase Bga1903, sourced from Burkholderia gladioli, which exhibited gliadin hydrolytic activity. The enzyme's secretion capability by Escherichia coli BL21(DE3) and its structural characteristics were previously elucidated through X-ray crystallography.
However, the present research sought to improve Bga1903's hydrolytic activity, specifically targeting the 33-mer and 26-mer peptides, by strategically modifying the S1 subsite through mutagenesis. Despite the efforts invested, the outcomes of the study appear to fall short of substantial progress. The alterations made to the S1 subsite were intended to enhance catalytic constants, but the text does not provide conclusive evidence or details about the degree of improvement achieved.
The critical review acknowledges the scientific effort invested in the study but highlights the lack of clear evidence regarding the effectiveness of the mutagenesis approach. The assessment suggests that further clarification and detailed results regarding the catalytic constants and overall impact of the modifications on Bga1903's hydrolytic activity are essential for a comprehensive understanding of the study's outcomes.
Author Response
Response: We appreciate the insight the reviewer provided us, and the revision was made according to the suggestions. The detailed information associated with the catalytic constants, including the saturation curve plots and double reciprocal plots of the substrate concentration dependence of the catalytic rate of Bga1903 and its variants, is added to the Appendix as Figures A1-A13. Compared to the wild-type Bga1903, mutant E380Q/S387L has a 39-fold increase in specific constant toward QPQ motifs, a common sequence that frequently appears in gluten-derived pro-immunogenic peptides. This trait is further exemplified by substantial hydrolytic activity against both 26-mer and 33-mer peptides.
Round 2
Reviewer 2 Report
Comments and Suggestions for Authors
In the revised manuscript that authors have attended all the points raised by this reviewer. This includes showing all the saturation curves from which kcat and Km values of wild type and mutants were derived, and the corresponding double reciprocal plots. However, upon showing these data, new important problems appear.
The Lineweaver-Burk plot is probably the worse transformation of saturation data to estimate kinetic parameters. This is particularly evident when all the used substrate concentrations are well below the Km. This is the case in most of the plots shown. In many cases, the Lineweaver-Burk line goes practically over the origin, a situation where very large differences of estimated parameters depend on very slight slope adjustment. The only really good solution to this problem is to increase the substrate concentration range tested. A less satisfactory solution would be to use other ways to estimate kcat and Km. You can use the Eadie-Hofstee plot (v versus v/S) or, even better, adjust the Michaelis-Menten equation to your data by non-linear regression analysis.
Independently of that, and also problematic, is that the plots represent substrate concentration as nmol. Obviously these are not concentration units, but just molar amount. These data cannot be converted by the reader to concentration because the volume of the assay reaction mixtures is unknown.
Km values are in mM. The plots should be also in mM.
Author Response
Please see the attached PDF file.

Reviewer 4 Report
Comments and Suggestions for Authors
Dear Authors,
Thank you for the revision
Acecpt in present form
Author Response
We thank you for your advice and patience
Round 3
Reviewer 2 Report
Comments and Suggestions for Authors
In this revised version of their manuscript, the authors have followed the suggestion to use the Eadie-Hofstee plot to derive Vmax and Km values. However, after doing this, the kcat and Km values reported in Table 1 and Table 2 are expressed as single values, rather than the means ± standard deviations shown in previous versions corresponsing to triplicate experiments. There is no reason for this loss of important information. From the triplicate experiments they must be able to derive triplicate values of kcat and Km obtained from Eadie-Hofstee plots.
Author Response
The means ± standard deviations of kcat and KM values, along with the explanations are now shown in Table 1 (Lines 124-131) and Table 2 (Lines 174-179). The numbers of kinetic constants within paragraphs 2.2 to 2.3 (Lines 111-162 and Lines 163-212) have also been adjusted accordingly. For visual clarity, only one set of triplicate experiments is utilized as the representative plots in the Appendix as Figure A1-A13 (Lines 429-507).